# Osmotic Fragility in Leukodepleted Stored Red Blood Cells: Implications for Neurocritical Care Transfusion Strategies

**DOI:** 10.3390/cells14211726

**Published:** 2025-11-03

**Authors:** Marta Peris, Maria A. Poca, Ana Ortuño, Verónica Pons, Nuria Rodríguez-Borrero, Desiree Jurado, Rafael Parra-López, Marina Rierola, Juan Sahuquillo

**Affiliations:** 1Neurotraumatology and Neurosurgery Research Unit, Vall d’Hebron Institut de Reserca (VHIR), Vall d’Hebron Barcelona Hospital Campus, Passeig de la Vall d’Hebron 119-129, 08035 Barcelona, Spain; marta.peris@vhir.org (M.P.); pocama@neurotrauma.net (M.A.P.); nuriarodriguez_2@hotmail.com (N.R.-B.); 2Department of Neurosurgery, Vall d’Hebron Hospital Universitari, Passeig de la Vall d’Hebron 119-129, 08035 Barcelona, Spain; 3Department of Surgery (Neurosurgery), Universitat Autònoma de Barcelona, 08193 Bellaterra, Spain; 4Erythropathology Unit, Service of Hematology, Vall d’Hebron Hospital Universitari, Passeig de la Vall d’Hebron 119-129, 08035 Barcelona, Spain; ana.ortuno.cabrero@gmail.com; 5Blood and Tissue Bank, Vall d’Hebron Hospital Universitari, Passeig de la Vall d’Hebron 119-129, 08035 Barcelona, Spain; vpons@bst.cat (V.P.); rparra@bst.cat (R.P.-L.); mrierola@bst.cat (M.R.); 6Neurotraumatology Intensive Care Unit, Vall d’Hebron Hospital Universitari, Passeig de la Vall d’Hebron 119-129, 08035 Barcelona, Spain; desiree.jurado@vallhebron.cat

**Keywords:** traumatic brain injury, red blood cell storage, osmotic fragility, free hemoglobin, transfusion

## Abstract

**Background**: Anemia is frequent in critically ill patients with traumatic brain injury (TBI) and worsens neurological outcomes. Red blood cell (RBC) transfusion is a cornerstone of management, but storage-related biochemical and structural changes may impair oxygen delivery. This study examined the effect of storage duration on osmotic fragility (OF) and free hemoglobin (fHb) in leukodepleted packed RBCs (pRBCs) as indicators of membrane stability and hemolysis. **Methods**: Twenty-four leukodepleted pRBC units in SAGM (saline, adenine, glucose, and mannitol) solution were analyzed from Day 3 to Day 42. OF was assessed by Beutler’s method with H50 values derived from logistic models, and fHb was quantified spectrophotometrically. Flow cytometry with phosphate-buffered saline (PBS)-induced osmotic stress provided complementary OF data. **Results**: OF increased significantly beyond 28 days, with Week 6 H_50_ values exceeding those at Weeks 2 and 4 (*p* < 0.0001). fHb rose progressively with storage: 7.3 ± 4.3 µmol/L (Week 2), 14.6 ± 7.9 (Week 4), and 25.7 ± 12.1 (Week 6) (*p* < 0.0001). Hemolysis remained below the 0.8% threshold but increased from 0.09% to 0.29% (*p* < 0.0001). **Conclusions**: pRBC storage beyond 28 days leads to greater OF and fHb release, reflecting reduced membrane stability. These changes may compromise transfusion efficacy and oxygen delivery in neurocritical care.

## 1. Introduction

Anemia is highly prevalent among critically ill patients, with approximately 40% of hospitalized individuals requiring at least one unit of packed red blood cells (pRBCs) during their hospital stay [1]. Approximately half of patients with moderate to severe traumatic brain injury (TBI) admitted to the intensive care unit (ICU) develop anemia, as documented in several studies [2,3,4]. The pathophysiology of anemia in TBI is complex and multifactorial, with contributing factors including acute blood loss, hemodilution, coagulopathy, and impaired erythropoiesis, among others. Traditional physiological principles, anecdotal evidence, and cohort studies suggest that anemia in patients with acute brain injuries may be detrimental, as it increases the risk of secondary brain damage and worsens neurological outcomes [2,3,4]. TBI patients with anemia typically experience prolonged hospitalizations, higher mortality rates, and poorer neurological outcomes compared with their non-anemic counterparts [4,5].

Management of anemia in patients with TBI often involves red blood cell (RBC) transfusions to enhance oxygen (O_2_) delivery and reduce the adverse effects of low hemoglobin (Hb) levels. Whole blood (WB) transfusions are still used in military medicine and low-income settings where the blood processing infrastructure is limited [6]. However, the standard of care in most medium- to high-income countries involves the transfusion of individual blood components, such as RBCs, plasma, and platelets. At most blood centers, donated WB is processed into RBC concentrates, which are subsequently leukoreduced through filtration to remove white blood cells and residual platelets [7]. This procedure is performed to reduce the risk of febrile non-hemolytic transfusion reactions, alloimmunization, and transmission of leukocyte-associated pathogens such as cytomegalovirus [8]. Leukodepleted RBCs are resuspended in an acidic additive solution at a hematocrit (Hct) of around 60% and stored in a refrigerated environment (1–6 °C) for a period of three to six weeks before transfusion [7]. The maximum permissible storage duration for pRBCs is 42 days in most European Union (EU) countries; nonetheless, some countries and institutions adopt more conservative policies. For instance, both the United Kingdom (UK) and the Netherlands limit pRBC storage to a maximum of 35 days [9,10].

The storage of pRBCs in blood banks is associated with a phenomenon known as the “storage lesion,” which refers [11] to a series of progressive metabolic, biochemical, and structural changes that occur in pRBCs over time during refrigerated storage [11]. These alterations include a decline in pH, reduced glycolytic activity, and depletion of key intracellular molecules, such as adenosine 5′-triphosphate (ATP) and 2,3-diphosphoglycerate (2,3-DPG), both of which are critical for RBC metabolism and O_2_ delivery. In addition, prolonged storage leads to the accumulation of extracellular potassium, the generation of reactive O_2_ species (e.g., superoxide radicals), and the release of free hemoglobin (fHb) into the supernatant. Collectively, these changes impair the structural integrity, deformability, and O_2_-carrying capacity of RBCs, potentially compromising their post-transfusion efficacy and safety. Some authors have compared storage lesions to the phenomena that occur during the aging of erythrocytes, which is accelerated by the storage conditions [12,13].

The suitability of older blood units for transfusion remains a concern, especially in relation to their diminished ability to transport and deliver O_2_ efficiently to peripheral tissues. Despite these concerns, the definition of what constitutes “fresh” versus “old” blood is inconsistent and lacks standardization, with no universally accepted storage duration threshold [14]. Collectively, storage lesions compromise the functionality and viability of stored pRBCs, potentially diminishing their efficacy and safety upon transfusion [15]. The increase in fHb reflects ongoing hemolysis in stored pRBC units [16]. As a potent scavenger of nitric oxide (NO)—the principal endogenous vasodilator—fHb may contribute to microvascular perfusion abnormalities following transfusion [14]. Additionally, the storage lesion involves oxidation injury to RBC, contributing to the formation of microparticles and the loss of their deformability [7,15,17]. While the rate at which the storage lesion occurs varies among donors, some data indicate that chronological age alone may not accurately predict the quality of stored blood [17].

The osmotic fragility (OF) of RBCs has been shown to increase with prolonged storage, potentially compromising the quality and safety of transfusions [12]. Hemolysis in stored pRBC units results from multiple factors that gradually impair membrane integrity, predisposing cells to rupture. Several studies have reported that RBCs stored for extended durations exhibit increased OF, mimicking the membrane instability seen in hereditary spherocytosis [13,18]. This increased fragility can result in accelerated post-transfusion clearance and reduced in vivo survival of transfused RBCs, ultimately compromising their therapeutic efficacy. As RBCs lose membrane integrity during storage, they tend to assume a more spherical morphology, rendering them more vulnerable to lysis even under modest osmotic stress [19]. Elevated OF is therefore a key contributor to the decline in the quality of stored blood, potentially compromising its effectiveness in transfusion therapy for anemic patients.

The objective of this study was to investigate the relationship between storage duration and OF in leukodepleted pRBCs, as well as to evaluate the effect of storage time on fHb levels in the supernatant. Both OF and fHb are considered surrogate markers of pRBC quality, yet their evolution during storage remains insufficiently characterized in the transfusion literature. We hypothesized that prolonged storage would increase OF and fHb, thereby compromising the ability of pRBCs to deliver oxygen effectively—a finding that, if confirmed, would be of particular relevance for defining transfusion strategies in anemic patients with acute brain injury.

## 2. Materials and Methods

This study was conducted as part of a broader translational research initiative led by the Neurosurgery and Neurotraumatology Research Unit at Vall d’Hebron University Hospital. The project comprises two components: (1) a laboratory investigation evaluating the biochemical and structural quality of stored pRBCs, and (2) an ongoing observational clinical study involving patients with moderate to severe TBI who received at least one unit of pRBCs [20]. The study was conducted in accordance with the Declaration of Helsinki, Spanish regulations, and EU directives, and was approved by the Ethics Committee of the Vall d’Hebron Research Institute (protocol identification: PR-AG-411/2017, approved 6 June 2017). All experimental studies were conducted in accordance with Spanish regulations and EU directives, and were approved by the Ethics Committee of the Vall d’Hebron Research Institute. The methodology and results of the in vitro studies to assess the OF of stored pRBC units are described in this paper.

### 2.1. Sample Size and Power Analysis

As this work was designed as an exploratory pilot study, the sample size (n = 24 leukodepleted pRBC units) was determined based on feasibility and alignment with previous pilot investigations that examined osmotic fragility (OF) and storage-related changes in stored red blood cells [21,22], which typically analyzed between 20 and 30 units.

The study included 24 leukodepleted packed red blood cell (pRBC) units, a sample size consistent with previous pilot studies assessing osmotic fragility (OF) and storage-related alterations in stored blood products [21,22]. To further verify the adequacy of this sample size, a post hoc power analysis was performed using G*Power v3.1.9.7 [23,24]. An F-test (ANOVA: repeated measures, within factors) was applied based on the observed variance and effect sizes obtained from Figure 6. For an effect size of 0.486, an α level of 0.05, and three groups (weeks), the calculated statistical power was 1.0, confirming that the sample size was sufficient to detect storage-related differences in OF across the three time points.

### 2.2. Leukodepleted Red Blood Cell Concentrates (RBC)

A total of 24 bags of leukodepleted pRBC units were obtained from the Blood and Tissue Bank of our institution. pRBC units were obtained in compliance with Regulation (EU) 2016/679 of the European Parliament and Council (27 April 2016) on the protection of personal data. Blood processing, from the moment of extraction in the Blood Bank until its analysis in the laboratory, followed these steps: WB (∼450 mL) was collected from healthy volunteer donors using CPD (citrate-phosphate-dextrose) as an anticoagulant-preservative solution. The leukocyte part and plasma of WB were removed by centrifugation so that only RBCs remained suspended in 100 mL of SAGM (saline, adenine, glucose, and mannitol) preservative solution. The pRBC units were obtained from the Blood Bank three days after collection, stored at 1–6 °C in our laboratory, and analyzed at intervals from day 3 to day 42 post-extraction.

### 2.3. Beutler’s Method for Assessing OF

To assess OF, we applied the method described by Beutler et al. [19], using varying sodium chloride (NaCl) solutions to induce different degrees of osmotic stress in twenty pRBC units. For the preparation of the dilution series, we used a stock solution containing 90 g NaCl (VWR Chemicals, Radnor, PA, USA), 13.95 g NaH_2_PO_4_ (Panreac, Chicago, IL, USA), and 2.34 g NaH_2_PO_4_·2H_2_O (Sigma, Burlington, MA, USA), diluted in 1000 mL Milli-Q water, and adjusted to pH 7.4. We then made dilutions equivalent to 9.0, 7.5, 6.5, 6.0, 5.5, 5.0, 4.0, 3.5, 3.0, 2.0, 1.0, and 0.0 g NaCl/L, with a calculated osmolality ranging from 0 mOsm/kg (distilled water) to 306.6 mOsm/kg (Table 1).

The pRBC units were maintained at 4 °C in our laboratory and examined at designated intervals between days 3 and 42 following collection. Two 50 µL blood aliquots were aseptically extracted from each unit and analyzed immediately at predefined time points: days 3, 9, 14, 18, 23, 28, 32, 37, and 42 post-collection. After extraction, samples were gently mixed with 5 mL of solutions containing the NaCl concentrations listed in Table 1. The suspensions were incubated at room temperature for 30 min, followed by centrifugation at 1200× *g* for 5 min. Hemolysis was evaluated by measuring the absorbance of Hb released into the supernatant. For this purpose, 200 µL of supernatant from each sample was transferred to a 96-well plate, and the optical density (OD) was measured at 540 nm using a SPECTROstar Nano spectrophotometer (BMG Labtech, Ortenberg, Germany). The mean OD of the two replicates for each RBC unit, corresponding to each saline concentration, was plotted to calculate the H_50_, defined as the NaCl concentration at which 50% hemolysis occurred. The OD value obtained from the supernatant following exposure to 0 g/L NaCl was used as the reference for 100% hemolysis.

### 2.4. Free Hemoglobin Accumulation and Storage-Induced Hemolysis

The quantification of fHb levels was performed using the Harboe direct spectrophotometric method, initially described in 1959 [25] and reviewed by Han et al. [26]. In this method, oxyhemoglobin is measured by detecting its absorbance peak around 415 nm, while also accounting for the absorbance of other plasma components (i.e., bilirubin/albumin complexes and lipids) at the same wavelength range. Harboe’s method uses a polychromatic formula by measuring the absorbance of samples at three different wavelengths: 380 nm (corresponding to the maximum absorbance of triglycerides), 415 nm, and 450 nm (corresponding to the maximum absorbance of bilirubin) [25,26]. The levels of fHb were determined with the following equation [22,23]:Hb (g/L) = (167.2 × A415 − 83.6 × A380 − 83.6 × A450)/1000.(1)

We analyzed twenty pRBC units obtained from twenty donors, stored from days 3 to 42 post-collection. Assessments were performed at predefined intervals: days 3, 7, 11, 14, 17, 21, 23, 25, 28, 30, 32, 35, 37, and 42. The samples underwent a two-step centrifugation process. Initially, 4 mL aliquots were extracted from the RBC units and centrifuged at 4000 rpm for 10 min using a Thermo Scientific SL 16R centrifuge (Thermo Fisher Scientific, Waltham, MA, USA). The resulting supernatant was then divided into 800 μL aliquots and further centrifuged in a Labnet Prism™ R centrifuge (LabNet Biotécnica, Madrid, Spain) at 15,000 rpm for 15 min. A 1:10 dilution of the supernatant from the second centrifugation was prepared in the cuvettes along with Milli-Q water. BRAND macro-cuvette cuvettes (Sigma-Aldrich, Burlington, MA, USA) were employed for analysis in the SPECTROstar Nano spectrophotometer (BMG Labtech, Ortenberg, Germany). The mean OD value of the two replicates for each RBC bag and time point was calculated to determine the fHb levels, using Harboe’s equation, and all measurements were multiplied by 10 and corrected for the 1/10 dilution of the sample in distilled water [27].

For the subsequent calculation of spontaneous hemolysis in the pRBCs, the following formula was used:Hemolysis (%) = [supernatant fHb (g/L) × (100 − Hct (%)] /tHb (g/dL)(2)
where fHb and tHb were free Hb and total Hb content [26]. The tHb and Hct measurements were determined by the GEM^®®^ PremierTM 4000 blood gas analyzer (Instrumentation Laboratory, Werfen, Bedford, MA, USA).

### 2.5. Flow Cytometry

To assess OF using an independent approach, we employed a flow cytometry (FCM) method previously described by Yamamoto et al. [28]. This technique evaluates RBC resistance to hemolysis under osmotic stress induced by varying concentrations of phosphate-buffered saline (PBS) solution [28]. OF was assessed by FCM in 19 pRBC units over a storage period of up to 42 days, with measurements obtained at the same predefined intervals used in Beutler’s method: days 3, 9, 14, 18, 23, 28, 32, 37, and 42 post-collection. From each RBC bag stored at 4 °C, 0.5 mL of blood was taken, rinsed with 2 mL of PBS (containing 1.37 M NaCl, 0.02 7M KCl, 0.08 M Na_2_HPO_4_, and 0.02 M KH_2_PO_4_, at room temperature and pH 7.4), and centrifuged at 1000 rpm for 5 min. The supernatant was discarded, and 2 μL of blood was resuspended in various PBS concentrations for 3 min. These concentrations were prepared by diluting a stock PBS solution with varying amounts of distilled water to obtain the target PBS percentages, as summarized in Table 1.

Sample analysis was conducted in a BD FACSCalibur flow cytometer (Becton-Dickinson, San José, CA, USA), and the subsequent analysis of the data obtained from the FCM was performed using CellQuest Pro software v5.1 (Becton Dickinson, San Jose, CA, USA). The number of events was set at 10,000, and we conducted the study of RBC size (FSC; forward scatter channels) and complexity (SSC; side scatter channels). The results obtained are represented in scatter plot diagrams as indicated by Yamamoto et al. [24]. In brief, the scatter plots were divided into four quadrants. The most relevant quadrants were the upper right, where the intact RBCs are represented due to their considerable size and complexity when exposed to 100% PBS (307.4 mOsm/kg) (Figure 1A) and the lower left quadrant, which reflects the accumulation of cell debris resulting from RBC hemolysis by subjecting the sample to high osmotic stress at 0% PBS (Figure 1B). To calculate the percentage hemolysis rate for each osmotic challenge, the following formula was used:1 − [cell count in the upper right quadrant/total cell count] × 100 (%)(3)
on the scatter plot provided by the FCM [28]. As in Beutler’s method, the hemolysis percentage data for each bag on each study day were plotted, and the H50 value was determined using the same methodology described previously.

### 2.6. Statistical Analysis

Descriptive statistics were calculated for each variable. For continuous variables, the assumption of normality was evaluated using skewness and kurtosis indices [29]. According to Kline’s criteria, skewness values ≤ 3.0 and kurtosis values ≤ 10 were considered indicative of a non-severe departure from normality [30]. The mean and standard deviation were used to describe continuous variables that followed a normal distribution, while the median, maximum, and minimum were reported for non-normally distributed data. Data are presented graphically using box-and-whisker plots. To compare differences in means across groups, a one-way repeated measures ANOVA was conducted, provided that the data did not show substantial deviations from normality and that Levene’s test confirmed the assumption of homogeneity of variances across groups [29]. If ANOVA assumptions were violated, the Kruskal–Wallis rank sum test was used. In the one-way ANOVA, an omnibus F-test was performed to identify overall differences among groups, and the effect size was estimated using eta squared (η^2^). If the omnibus F-test yielded statistical significance, post hoc pairwise comparisons between groups were performed using the Bonferroni adjustment to control for the risk of Type I error due to multiple comparisons [29].

For evaluating the OF at different storage days, we used the H_50_ calculated by the R *drc* package [31]. In brief, for each group of bags, we used the function in *drc* package to fit the four-parameter log-logistic function and obtain the Hill coefficient and the ED_50_ (H_50_) [31]. The threshold for statistical significance was considered at *p* < 0.05. Statistical analyses were carried out with R v4.5.0 [32] and the integrated development environment R Studio v2024.12.1 (RStudio, Inc., Boston, MA, USA) using the *tidyverse*, *rstatix*, and *psych* packages [33,34].

**Data management:** This study was conducted according to the FAIR principles committed to making data and services Findable, Accessible, Interoperable, and Reusable (https://www.go-fair.org, last accessed: 23 July 2025). The entire anonymized dataset, metadata, and dictionary are available for download from https://zenodo.org (DOI:10.5281/zenodo.17086161). Given the descriptive and exploratory design of this study, a formal power analysis was not performed. Instead, the sample size (n = 24 units) was determined pragmatically to ensure feasibility while remaining consistent with sample sizes commonly employed in comparable descriptive studies in the transfusion literature.

## 3. Results

### 3.1. Osmotic Fragility: Beutler’s Method

Beutler’s method was employed to assess RBC OF under conditions of graded osmotic stress, as detailed in the Section 2. An initial exploratory data analysis was conducted by plotting the mean hemolytic dose (H_50_), defined as the NaCl concentration required to induce 50% hemolysis, against the duration of RBC storage. This approach enabled a visual and statistical evaluation of the temporal evolution of RBC membrane stability over the storage period. Of the 180 measurements, only one data point was missing on day 32 post-collection; it was imputed using the median H_50_. Boxplots in Figure 2 clearly showed two populations of RBCs defined by the difference in means around the grand mean of the whole population, which was 4.60 g/L NaCl (156.8 mOsm/kg). To evaluate differences in H_50_ means over storage days, we used one-way ANOVA for repeated measures, as skewness and kurtosis values for all storage days did not show any significant departure from a normal distribution in any of the nine storage groups (20 bags per group). The mean H_50_ values on days 32, 37, and 42 differed significantly from the mean H_50_ values observed between days 3 and 28. The omnibus ANOVA was significant: F(8144) = 16.6, *p* < 0.0001, η^2^ = 0.36.

To improve clarity and facilitate statistical analysis, the 180 OF measurements were grouped into three storage age categories: Week 2 (days 3, 9, and 14), Week 4 (days 18, 23, and 28), and Week 6 (days 32, 37, and 42). Figure 3 presents box plots of H_50_ values across the three defined storage age groups, each comprising 60 measurements. A progressive increase in OF was observed with extended storage duration. In Week 2 (days 3–14), the median H_50_ was 4.52 g/L NaCl, corresponding to an osmotic stress of approximately 154 mOsm/kg. By Week 4 (days 18–28), a slight elevation in H_50_ was noted (median: 4.56 g/L; range: 3.73–4.93 g/L). In Week 6 (days 32–42), a marked increase in H_50_ was evident (median: 4.83 g/L; range: 3.88–5.73 g/L), indicating greater susceptibility to hemolysis. At this stage, 50% of RBCs were lysed when exposed to a medium with an osmotic stress equivalent to approximately 164.6 mOsm/kg, suggesting reduced membrane stability in older stored units. A repeated-measures one-way ANOVA was conducted to evaluate the effect of storage duration on pRBC OF, measured by H_50_. The analysis revealed a significant main effect of storage week (F(2.38) = 23.8, *p* < 0.000001), with a large effect size (η^2^ = 0.56), indicating a substantial increase in OF as storage duration progressed. Post hoc paired t-tests with Bonferroni correction revealed no significant difference in H_50_ values between Week 2 and Week 4 (*p* = 0.20). In contrast, H_50_ values were significantly higher in Week 6 than in both Week 2 (*p* < 0.000001) and Week 4 (*p* < 0.000001), reflecting a pronounced decline in membrane integrity beyond 28 days of storage.

As an additional confirmatory analysis, dose–response curves were constructed for each storage group (Weeks 2, 4, and 6) using the drc package in R [31]. In the context of OF testing, “dose” corresponds to the saline concentration—where lower concentrations impose greater osmotic stress—and “response” represents the percentage of hemolyzed RBCs. For each storage group, a four-parameter log-logistic model was fitted to the data. From these models, the saline concentrations required to induce 50% hemolysis (H_50_) were estimated, along with the slope of the curve at the H_50_ point, which reflects the steepness of the hemolytic response and the rate of membrane failure.

The three dose–response curves are presented in Figure 4. In this analysis, the dose–response curves for Week 2 and Week 4 exhibited substantial overlap. Nonetheless, a small but statistically significant difference was detected between the two curves within the 50–80% hemolysis range (*p* = 0.0012). The most pronounced differences, however, were observed in comparisons of H_50_ values between Week 2 and Week 6 (*p* < 0.0001), as well as between Week 4 and Week 6 (*p* < 0.0001). These findings confirm that OF remains relatively stable when the storage duration of RBC does not exceed 28 days post-extraction, but increases significantly with prolonged storage beyond this time.

### 3.2. Osmotic Fragility: Flow Cytometry

Using this second method, the number of hemolyzed cells was quantified at each PBS gradient concentration, and the H_50_ was calculated as the percentage of PBS inducing 50% hemolysis. The OF of 19 pRBC units was analyzed over a storage period of up to 42 days. To ensure methodological consistency with the previous experiment using Beutler’s method, the pRBC units were similarly categorized into three storage time groups: Week 2, Week 4, and Week 6. This approach facilitated direct comparison of OF measurements across both methodologies and storage durations. For each storage week, a four-parameter log-logistic model was applied consistently to estimate the PBS concentrations required to induce 50% hemolysis (H_50_). This standardized modeling approach ensured comparability across the three storage groups. The resulting dose–response curves are presented in Figure 5.

Using flow cytometry, the dose–response curves for Week 2 and Week 4 showed substantial overlap, with H_50_ values of 47.3% (145.4 mOsm/kg) and 49.0% (150.6 mOsm/kg), respectively. No statistically significant difference was observed between these two time points (t = −2.06; *p* = 0.054). In contrast, H_50_ values differed significantly between Week 2 and Week 6 (*p* < 0.0001) and between Week 4 and Week 6 (*p* = 0.0015). These results reinforce the finding that OF remains relatively stable during the first 28 days of storage, but increases significantly with prolonged storage beyond this threshold.

### 3.3. Free Hemoglobin in the Stored Bags

fHb levels were analyzed using the same strategy applied to the OF data. All bags and measurements were grouped into the three predefined categories based on storage duration: Week 2, Week 4, and Week 6. This approach ensured consistency across analyses and facilitated direct comparisons of fHb levels over time. Summary data for fHb levels across the three storage groups (Weeks 2, 4, and 6) are presented as box-and-whisker plots in Figure 6.

Of the total dataset (n = 336), 12 missing measurements were imputed using the median value to preserve the integrity of the analysis. Mean fHb was 7.29 ± 4.34 µmol/L in the Week 2 group, 14.57 ± 7.91 µmol/L in the Week 4 group, and 25.73 ± 12.09 µmol/L in the Week 6 group. Skewness was below 1.8 in all groups, and a maximum kurtosis of 3.55 was found for measurements in Week 2. Despite the non-severe departure from normality of the distribution of fHb for each level of the grouping variable, Levene’s test did show that the variance of fHb for each level of the independent variable was not homogeneous: F(2321) = 34.3, *p* < 0.001. Because of this, we used the alternative Welch one-way test to compare the mean fHb without assuming equal variances for all groups. The results of the omnibus ANOVA indicated a significant effect of storage time on fHb levels: F(2197) = 130, *p* < 0.0001. We followed up the significant omnibus with a series of post hoc, pairwise comparisons controlling for Type I error with the traditional Bonferroni adjustment. Results suggested that there were statistically significant differences between the three groups with different storage time points (*p* < 0.0001 for all three groups).

The percentage of hemolyzed RBCs in all bags analyzed remained below the recommended threshold of 0.8%. In units stored for 3 to 14 days (n = 96), hemolysis was negligible, with a median of 0.09% (range: 0.03–0.30). Hemolysis increased to a median of 0.17% (range: 0.07–0.76) in pRBC units stored for 17 to 29 days (Week 4), and further increased to a median of 0.29% (range: 0.08–0.66) in units stored beyond 28 days (Week 6).

## 4. Discussion

Effective management of TBI focuses on preventing and mitigating secondary brain injuries, which often develop within hours to days following the initial insult. Among these contributing factors, anemia is frequently observed in TBI patients and has been associated with unfavorable neurological outcomes [4,5,35]. In their seminal 1978 study, Miller et al. identified anemia—together with hypoxia, hypercapnia, and hypotension—as a modifiable early systemic insult strongly associated with increased morbidity and mortality in patients with severe TBI [36].

Anemia is highly prevalent among critically ill patients and typically develops early during their stay in the ICU. By the third day of admission, nearly 95% of ICU patients exhibit some degree of anemia [37]. In the context of TBI, anemia has been consistently associated with poorer neurological outcomes, prolonged hospitalization, and increased mortality compared with non-anemic TBI patients [5,35]. Despite its high prevalence, the clinical management of anemia—particularly the decision to initiate RBC transfusion—remains a subject of ongoing debate. While transfusion may enhance cerebral O_2_ delivery and potentially mitigate hypoxic injury, it has also been associated with adverse outcomes. These observations raise significant concerns about the routine use of transfusion in neurocritical care, particularly when decisions are based on rigid Hb thresholds rather than individualized physiological parameters. Furthermore, systematic reviews in the TBI literature have failed to demonstrate a consistent clinical benefit associated with RBC transfusion, underscoring the persistent uncertainty regarding its therapeutic value and the delicate risk-benefit balance of transfusion strategies in this vulnerable population [38].

The ongoing debate over ‘permissive anemia’ as a strategy in neurocritical care began with the publication of a pivotal clinical trial by Hebert et al. in 1999 [39]. The Transfusion Requirements in Critical Care (TRICC) trial supported the adoption of restrictive transfusion thresholds in critically ill adults without active bleeding. It concluded that a restrictive strategy—typically maintaining hemoglobin levels between 7–8 g/dL—was as safe and effective as a more liberal approach targeting 9–10 g/dL [39]. Since the publication of this influential trial, restrictive transfusion strategies have been validated as safe across broader ICU populations. However, their applicability to neurocritical care patients remains controversial.

The Collaborative European NeuroTrauma Effectiveness Research in Traumatic Brain Injury (CENTER-TBI) study collected comprehensive data on injury characteristics, clinical management, and outcomes of TBI patients across multiple European centers [40]. Recent data from the CENTER-TBI cohort suggest that transfusion decision-making in TBI—a prototypical scenario in neurocritical care—is highly variable and often guided by individual clinician preference rather than standardized thresholds [4]. This variability reflects the discrepancies observed in major clinical trials, underscoring the need for further targeted research to determine optimal transfusion strategies in neurocritical populations, where the physiological demands of the injured brain may render conventional ICU thresholds inadequate.

The debate over the “permissive anemia” strategy in acute brain injury has intensified following the publication of three major clinical trials in 2024—TRAIN, HEMOTION, and SAHaRA—each reaching different conclusions. The multicenter TRAIN trial, which enrolled a heterogeneous cohort of patients with acute brain injury—including TBI, aneurysmal subarachnoid hemorrhage (aSAH), and intracerebral hemorrhage—demonstrated that a liberal transfusion strategy (hemoglobin threshold of 9 g/dL) was associated with a significantly lower rate of unfavorable neurological outcomes at 180 days compared with a restrictive strategy (7 g/dL) (adjusted relative risk = 0.86; 95% CI, 0.79–0.94; *p* = 0.002) [41]. Patients in the liberal group also experienced fewer cerebral ischemic events, supporting the potential benefit of maintaining higher hemoglobin concentrations in this clinical context. By contrast, in patients with isolated TBI, the HEMOTION trial—which compared liberal (Hb threshold 10 g/dL) and restrictive (7 g/dL) transfusion strategies—found no significant difference in the incidence of unfavorable neurological outcomes at six months between the two groups [42]. Similarly, the SAHaRA randomized controlled trial (Aneurysmal SubArachnoid Hemorrhage–Red Blood Cell Transfusion and Outcome) compared a liberal strategy (Hb threshold 10 g/dL) with a restrictive approach (Hb threshold 8 g/dL) in patients with aSAH and anemia. The study did not demonstrate a significant difference in the risk of unfavorable neurological outcomes at 12 months between the liberal and restrictive transfusion groups [43].

Reflecting this heterogeneity, general ICU guidelines continue to support restrictive transfusion thresholds (typically 7–8 g/dL) for most critically ill adults, as recommended in recent reviews [44]. In contrast, disease-specific guidelines, such as the 2023 American Heart Association/American Stroke Association (AHA/ASA) statement on aneurysmal aSAH, refrain from endorsing a fixed Hb threshold. Instead, they advocate for individualized transfusion decisions informed by the clinical context and multimodal neuromonitoring, including intracranial pressure (ICP), cerebral perfusion pressure (CPP), and brain tissue oxygen pressure (PbtO_2_) [45]. Overall, while liberal transfusion strategies may offer some benefit in heterogeneous cohorts of acute brain injury, recent data suggest neutral outcomes in isolated TBI and aSAH.

In a comprehensive review of all major published transfusion trials—including the most recent studies cited above—an important caveat must be acknowledged: the thresholds used to define anemia are somewhat arbitrary and often diverge from internationally recognized standards. For instance, the World Health Organization (WHO) defines anemia as an Hb concentration below 13 g/dL in men and below 12 g/dL in non-pregnant women [46]. However, most clinical trials in neurocritical care adopt significantly lower thresholds to define anemia and guide transfusion strategies. This discrepancy poses a fundamental limitation in both the interpretation and clinical applicability of transfusion trial results. Patients classified as ‘non-anemic’ under trial-specific definitions may, in fact, fall below the WHO-defined normal range—thereby underestimating the physiological burden of suboptimal Hb levels, particularly in vulnerable populations. A clear example is provided by the secondary analysis of the CENTER-TBI cohort, where Guglielmi et al. defined anemia using a threshold of <9.5 g/dL [4]. This value diverges significantly from WHO standards and may be insufficient to ensure adequate cerebral O_2_ delivery in patients with TBI, especially those with impaired autoregulatory capacity or elevated intracranial pressure—both common in the neurocritical care setting.

### 4.1. RBC Storage Lesion: A Missing Link in Transfusion Outcomes

A second important caveat in most multicenter transfusion trials is the lack of objective data regarding the quality and storage duration of the RBCs administered. In the majority of these studies, neither the chronological age of the transfused units nor critical indicators of the so-called “storage lesion”—such as fHb levels, OF, ATP depletion, or hemolysis indices—are systematically documented. This omission introduces a potentially confounding variable that may influence the interpretation of trial outcomes.

RBCs comprise approximately 40% to 45% of total blood volume and are the primary mediators of gas exchange, transporting O_2_ and carbon dioxide (CO_2_) through Hb [47]. RBCs are derived from pluripotent hematopoietic stem cells in the bone marrow and mature under the regulation of erythropoietin, a glycoprotein hormone primarily synthesized in the kidneys [47]. During erythropoiesis, RBCs undergo terminal differentiation, losing their nuclei and organelles and acquiring a highly deformable, biconcave shape. Mature RBCs have a circulating lifespan of approximately 100–120 days, after which they are cleared by antigen-presenting cells, particularly dendritic cells [12].

However, these properties are progressively compromised during ex vivo storage, resulting in a series of structural and metabolic changes collectively referred to as the “storage lesion”. The storage lesion encompasses oxidative membrane damage, decreased deformability, microvesicle formation, and hemolysis—all of which contribute to reduced post-transfusion survival of RBCs and diminished capacity for effective O_2_ delivery to tissues. OF testing offers a functional assessment of RBC membrane integrity and serves as a valuable biomarker for evaluating the quality of stored RBCs. In clinical settings where optimizing tissue oxygenation is paramount—such as in TBI—OF may provide critical information about the appropriateness of stored RBCs for transfusion.

Harboe’s spectrophotometric method employed in our study to assess OF remains the gold standard for quantifying fHb. Despite its analytical robustness, this method has yet to be fully automated for routine clinical application, primarily due to its reliance on skilled personnel and the need for wavelength-specific absorbance measurements [27]. In our analysis, spontaneous hemolysis increased by approximately 33% on day 42 compared with day 3 post-extraction. While all measured values remained below internationally accepted regulatory thresholds—0.8% set by the European Council and 1% by the U.S. Food and Drug Administration [48]—the observed progressive rise in fHb is indicative of cumulative damage to the RBC membrane occurring during storage [7,15].

In our study, we observed a significant increase in OF in leukodepleted pRBC units after 28 days of storage. While OF remained relatively stable during the first four weeks, a marked rise in H_50_ values and shifts in the slope of the hemolysis curve were detected between days 32 and 42, indicating progressive membrane destabilization. These findings are consistent with those reported by Mustafa et al., who also demonstrated that OF increases with prolonged storage of pRBCs using a similar storage protocol [21]. In their study, the progressive rightward shift in the OF curve—illustrated in their Figure 1—suggested that RBCs stored for 42 days exhibited markedly reduced resistance to hypotonic stress, a hallmark of advanced membrane fragility [21].

Morphological alterations of stored RBCs, including the progressive emergence of echinocytes, stomatocytes, and spherocytes (Figure 7, unpublished results), are well-recognized hallmarks of the storage lesion. These shape abnormalities compromise RBC deformability and impair microvascular transit, thereby reducing the efficiency of tissue O_2_ delivery. In parallel, the observed increase in OF—validated in our study through both spectrophotometric and FCM methodologies—is consistent with previous findings indicating that prolonged storage reduces the mechanical resilience and deformability of RBCs [22]. The progressive accumulation of fHb in the supernatant of stored RBC units further reinforces this trend. None of the units exceeded the regulatory threshold for hemolysis (0.8% as defined by the European Council); however, fHb levels rose markedly between Weeks 4 and 6, reflecting cumulative membrane damage and ongoing subclinical hemolysis during extended storage.

As fHb is a potent scavenger of NO—a key endogenous vasodilator—its accumulation and subsequent transfusion may contribute to microvascular dysfunction. This is particularly relevant in patients with impaired cerebral autoregulation, such as those with TBI. In our study, we observed progressive morphological changes in stored pRBC units (Figure 7, unpublished results), which we hypothesize are mediated by oxidative damage to the RBC cytoskeleton. This structural compromise likely promotes membrane vesiculation and the emergence of spherocytes, a morphologically rigid population of RBCs characterized by increased OF and reduced deformability, as previously reported by Mustafa et al. [21]. Such structural changes impair the ability of RBCs to traverse microcirculation and may shorten their lifespan post-transfusion, ultimately compromising their therapeutic efficacy.

### 4.2. OF, Storage Lesions, and the Transfusion Dilemma

Our findings carry important clinical implications for neurocritical care. In patients with TBI and other forms of acute brain injury, even subtle impairments in cerebral O_2_ delivery can adversely influence neurological recovery. The present data suggest that leukodepleted pRBCs stored beyond 28 days exhibit measurable deterioration in membrane integrity and increased OF, potentially compromising their capacity to restore tissue oxygenation effectively. Despite this, current transfusion protocol and traditional blood bank practices primarily rely on visual inspection, unit age (up to 42 days per EU guidelines [49]), and standard hemolysis thresholds (e.g., <0.8% in the EU and <1% in the United States [49,50]) to determine the acceptability of pRBC units for clinical use.

However, these conventional parameters may not fully capture the functional quality of stored RBCs, particularly in high-risk populations such as neurocritical patients. Our findings support the notion that additional indicators—such as OF, spontaneous hemolysis, or other storage lesion markers—should be considered when evaluating the suitability of pRBCs for transfusion. We propose that more selective transfusion strategies, incorporating both storage age and objective quality metrics, may be warranted in vulnerable populations where O_2_ delivery is tightly coupled to clinical outcomes. Future studies should aim to define thresholds for these supplementary parameters and validate their predictive value in prospective clinical settings.

Our results underscore the need to consider not only transfusion thresholds but also the storage-related quality of RBC units, as these factors may directly influence cerebral oxygenation and clinical outcomes. Given the evidence that older blood may exhibit diminished deformability, increased membrane fragility, and impaired O_2_ delivery capacity, a more granular characterization of transfused blood quality is essential in future clinical trials, especially when transfusion efficacy is a central clinical concern. Together, the fHb and OF data highlight the potential value of combining biochemical and biophysical assays to better evaluate RBC integrity during storage and refine transfusion strategies.

Some clinical studies have reported no significant differences in outcomes between transfusions involving “fresh” versus older stored blood; nevertheless, our findings reinforce the utility of OF as a relevant biomarker of storage-related RBC deterioration. OF provides information about the structural integrity of the RBC membrane, which becomes progressively compromised during storage. These alterations are secondary effects of therapeutic interventions aimed at controlling elevated ICP, such as the administration of hyperosmolar agents like mannitol or hypertonic saline. As a result, both hyponatremia and hypernatremia are frequently observed in this population.

### 4.3. Reevaluating Storage Duration and Quality Control in Transfusion Practice

While fluctuations in serum osmolality are generally well tolerated by healthy RBCs, they may precipitate hemolysis in fragile storage-damaged erythrocytes. This is especially pertinent in neurocritical care, where patients frequently experience marked osmotic shifts due to the therapeutic use of hyperosmolar agents such as mannitol or hypertonic saline for ICP management. Both hyponatremia and hypernatremia are commonly observed in this context. Manrique-Castaño et al. reported a striking case of severe hemolysis in a patient with hereditary spherocytosis/elliptocytosis who developed profound hyponatremia (114 mmol/L) secondary to Sheehan syndrome [51]. OF testing in this patient revealed a pronounced rightward shift in the OF curve, with an estimated H_50_ at a NaCl concentration of approximately 6.7 g/L (Figure 2 of the paper), suggesting marked membrane fragility. This case indeed involved a congenital membrane pathology, yet it still underscores a broader principle: RBCs with compromised membrane integrity—whether hereditary or storage-induced—are more susceptible to osmotic stress and may hemolyze under conditions otherwise tolerable to healthy cells. If confirmed by larger-scale studies, our findings suggest that the quality of transfused blood—particularly after 28–30 days of storage—may be sufficiently compromised to impair its therapeutic efficacy, particularly in terms of O_2_ delivery. Such degradation could have clinically relevant consequences, particularly in patients with acute brain injuries, where even minor deficits in tissue oxygenation may worsen outcomes.

These results highlight the need to reassess current transfusion practices and support the integration of standardized quality control protocols into blood bank operations.

Specifically, OF testing should be considered a valuable functional biomarker for the routine evaluation of stored pRBC units, thereby enabling a better definition of the optimal storage duration for safe and effective transfusion. While the Beutler method remains the reference standard for OF assessment [19], alternative techniques based on FCM—such as the method described by Yamamoto et al. [28]—offer significant practical advantages. Their protocol employs pre-prepared PBS dilutions, facilitating rapid, reproducible, and operator-independent analysis. In our experience, the use of PBS solutions simplifies and accelerates the preparation of osmotic gradients. Taken together, these trials highlight the persistent uncertainty regarding the optimal transfusion threshold in patients with acute brain injury. While the TRAIN trial suggests a potential benefit of a liberal transfusion strategy in a heterogeneous cohort, the HEMOTION and SAHaRA trials did not confirm such advantages in more narrowly defined populations. These discrepancies likely reflect variations in patient characteristics, underlying pathophysiology, and the differential impact of anemia on cerebral oxygenation across distinct subtypes of brain injury. Collectively, these findings emphasize the importance of adopting individualized transfusion strategies in neurocritical care—approaches that consider not only Hb concentrations but also the quality of transfused RBCs and the unique physiological vulnerabilities of the injured brain.

### 4.4. Study Limitations

RBCs undergo structural and functional changes during storage that compromise their mechanical resilience. Although our study did not directly assess mechanical fragility, evidence shows that alterations in cell morphology and membrane properties closely mirror their resilience to stress. OF testing provides valuable information, as cells that are more prone to lysis under hypotonic conditions typically show reduced deformability and greater susceptibility to mechanical damage. Berezina et al. demonstrated that RBCs losing their normal discocyte morphology during storage show decreased flexibility and increased hemolysis rates [52]. Similarly, RBCs from patients with conditions such as renal failure, diabetes, sepsis, or acute inflammation often exhibit more spherical shapes accompanied by impaired deformability, supporting a strong connection between morphology, OF, and mechanical strength [53].

Recent advances in microfluidic technologies offer promising new avenues to detect sub-lethal damage in stored RBCs with high sensitivity and throughput. Unlike conventional OF assays, microfluidic platforms can assess subtle biophysical changes at the single-cell level, including alterations in deformability, membrane stability, and flow dynamics under physiologically relevant shear stress [54,55]. Several groups have developed lab-on-a-chip systems that enable real-time monitoring of RBC passage through microchannels mimicking the capillary network, allowing early detection of storage-induced impairments before overt hemolysis occurs [55]. Our study has several limitations that should be taken into consideration. First, as an in vitro investigation, it does not provide direct evidence regarding in vivo RBC survival, microvascular function, or clinical efficacy following transfusion. First, our sample size was adequate to detect significant storage-related alterations in OF and fHb; nonetheless, the clinical relevance of these findings remains to be confirmed in well-designed patient studies. Second, our analysis was limited to leukodepleted pRBCs, and it is unclear whether similar patterns of membrane destabilization would be observed in non-leukodepleted units or under different storage protocols. Third, we did not assess post-transfusion outcomes, such as RBC lifespan or O_2_ delivery efficacy, which are critical for determining the real-world impact of storage-induced changes. Future research should aim to integrate in vitro markers—such as OF and fHb—with clinical endpoints in neurocritical care settings to establish evidence-based thresholds for storage duration and transfusion quality.

### 4.5. Clinical Implications and Future Directions

Our findings raise important considerations for transfusion practices in neurocritical care. Patients with TBI and other acute brain insults are especially vulnerable to secondary injuries related to impaired O_2_ delivery. In this setting, even subtle declines in the quality of stored RBCs may compromise therapeutic efficacy. The observed increase in OF and fHb after 28–30 days of storage suggests that transfused pRBCs may lose functional capacity well before the currently accepted 42-day storage limit.

Despite this, contemporary transfusion protocols rarely incorporate storage duration or RBC quality markers into clinical decision-making. Our data support the hypothesis that incorporating OF testing into blood bank workflows may offer added value in identifying units more suitable for patients at high risk of ischemic injury—such as those with TBI or elevated ICP.

Future studies should validate these findings in prospective clinical trials, linking in vitro parameters to functional outcomes—such as cerebral oxygenation, transfusion efficacy, and neurological recovery. Additionally, research should explore whether modified storage protocols or targeted blood selection strategies may improve outcomes in vulnerable populations. Until such data are available, transfusion strategies in neurocritical care should be individualized, incorporating both physiological thresholds and qualitative assessments of blood product integrity.

## 5. Conclusions

This study highlights the progressive deterioration in RBC membrane stability, as reflected by increased OF and fHb, during prolonged storage of leukodepleted pRBCs. These alterations become particularly evident after 28–30 days of storage and may have critical implications for transfusion efficacy in neurocritical care populations. In conditions such as TBI, where optimal O_2_ delivery is essential and autoregulatory mechanisms may be impaired, the transfusion of functionally compromised RBCs could limit the therapeutic benefit or even contribute to adverse outcomes.

Our results support considering OF as a practical biomarker of RBC quality and underscore the need to refine current transfusion protocols. Integrating standardized, rapid OF testing into routine blood bank practices may improve the selection of RBC units for high-risk patients. Ultimately, the combination of in vitro quality assessment and individualized clinical criteria may lead to safer and more effective transfusion strategies in neurocritical care. Further research is warranted to validate these findings in clinical settings and to establish evidence-based thresholds that optimize both safety and therapeutic efficacy.

### Keypoints

RBC OF increases significantly after 28 days of storage, indicating a progressive decline in membrane integrity and heightened susceptibility to hemolysis. These alterations may adversely affect post-transfusion RBC survival and impair O_2_ delivery—particularly relevant in neurocritical care patients, where optimal tissue oxygenation is essential.OF testing using flow cytometry provides a rapid and reliable method to assess the quality of stored pRBCs. This technique has the potential to serve as a point-of-care tool, enabling the personalization of transfusion strategies based on RBC integrity.Incorporating storage duration into transfusion decision-making may help improve clinical outcomes in patients with acute brain injuries, where the efficacy of O_2_ delivery is critical to neurological recovery.Further clinical studies are needed to evaluate the impact of the storage time and quality of RBCs on long-term neurological outcomes and to determine evidence-based thresholds for transfusion in neurocritical populations.

## Figures and Tables

**Figure 1 cells-14-01726-f001:**
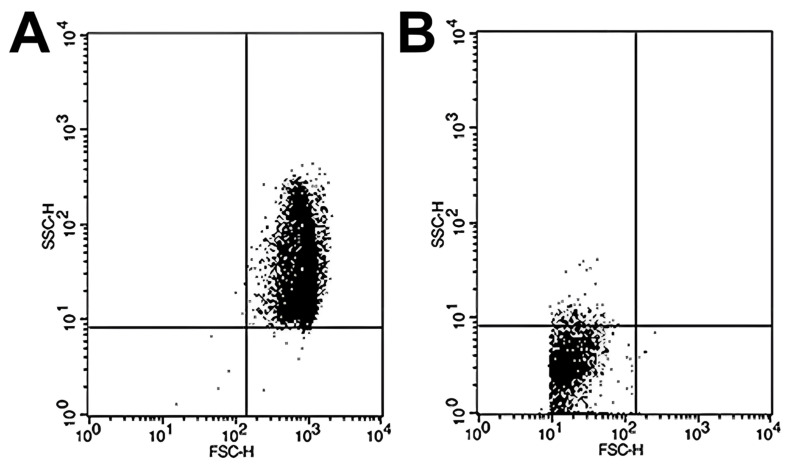
Representative scatter plots of RBC samples analyzed by flow cytometry under different osmotic conditions. Flow cytometry analysis was performed to assess RBC osmotic fragility by exposing samples to varying PBS concentrations. (**A**) At 100% PBS, the majority of events appear in the upper right quadrant of the scatter plot, corresponding to intact, non-hemolyzed RBCs characterized by preserved size and internal complexity. (**B**) At 0% PBS (distilled water), most events shift to the lower left quadrant, representing cellular debris resulting from complete hemolysis and associated with markedly reduced forward and side scatter signals, indicative of diminished cell size and structural complexity.

**Figure 2 cells-14-01726-f002:**
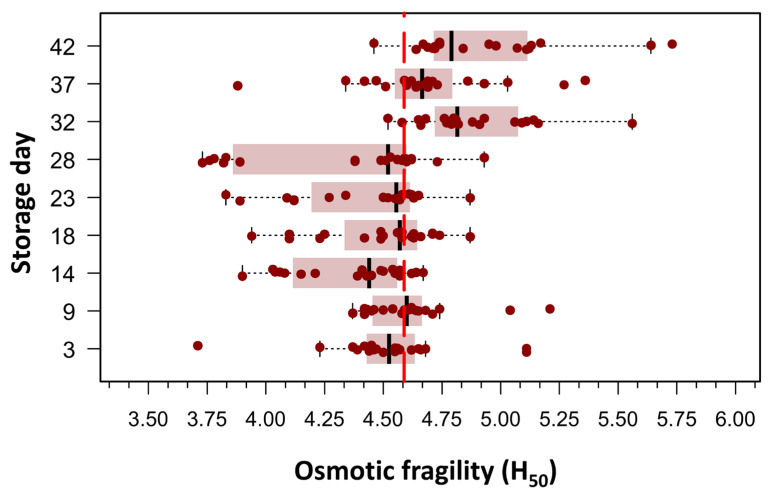
Box plots illustrating the concentration required to induce 50% hemolysis (H_50_) across different storage times, as evaluated using spectrophotometry (Beutler’s method). Each storage day represents the H_50_ of 20 pRBC concentrates, measured from day 3 post-extraction up to day 42. Outliers (n = 11), defined as values above Q1 + 1.5 × IQR or below Q1 − 1.5 × IQR, where Q1 and Q3 are the first and third quartiles and IQR represents the interquartile range, were excluded. The figure reveals two distinct RBC populations, distinguished by differences in their mean H_50_ values relative to the grand mean of the entire dataset (4.60 g/L NaCl; 156.8 mOsm/kg). As this was an exploratory analysis aimed at visualizing population distribution, no formal statistical testing was performed for this figure.

**Figure 3 cells-14-01726-f003:**
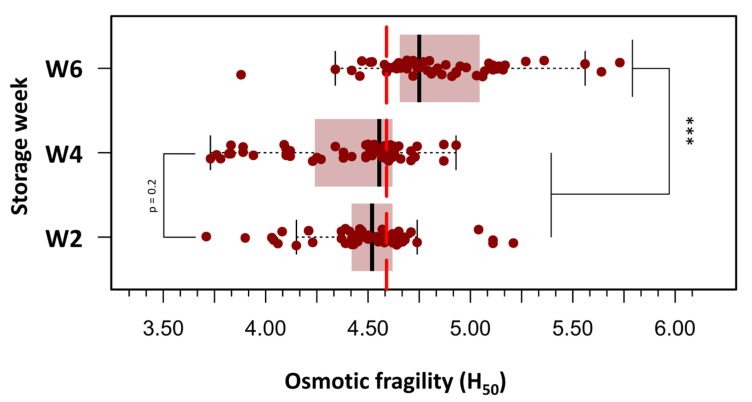
Box plots depicting the distribution of H_50_ values across three storage intervals: **Week 2** (days 3–14), **Week 4** (days 18–28), and **Week 6** (days 32–42) post-collection. A progressive increase in osmotic fragility was observed with storage duration, with statistically significant differences noted between Week 6 and both earlier time points. Boxes represent the interquartile range (IQR), horizontal lines indicate the median, and whiskers extend to 1.5 × IQR. Individual outliers are displayed as separate points. Statistical analysis was performed using repeated measures ANOVA followed by Bonferroni-adjusted paired t-tests. (***) *p* < 0.0001.

**Figure 4 cells-14-01726-f004:**
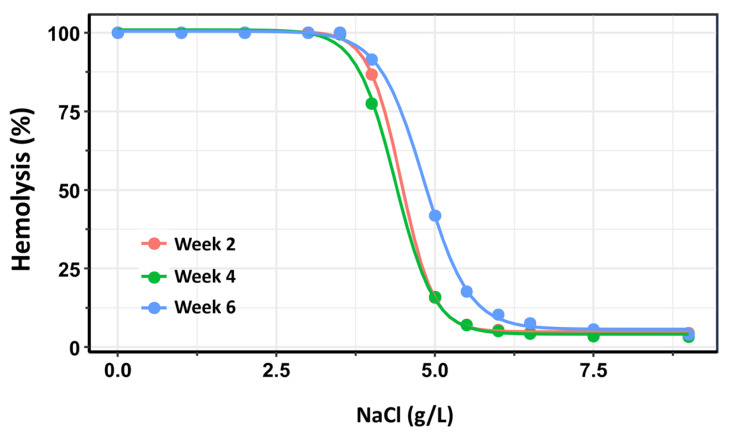
Dose–response curves depicting the percentage of hemolysis as a function of NaCl concentration across three storage intervals: Week 2 (pink), Week 4 (green), and Week 6 (blue) post-extraction. The rightward shift in the curves and changes in their slopes over time reflect a progressive increase in osmotic fragility with prolonged storage duration. These trends indicate reduced membrane stability and increased susceptibility to lysis of red blood cells stored beyond four weeks.

**Figure 5 cells-14-01726-f005:**
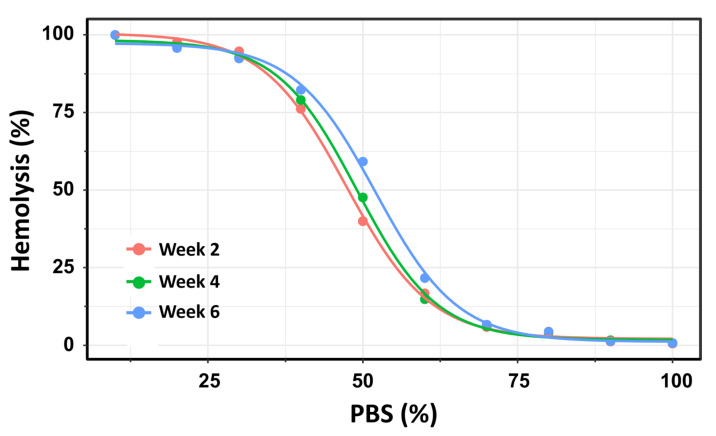
Dose–response curves depicting the percentage of hemolysis as a function of PBS concentration across three storage intervals: **Week 2** (pink), Week 4 (green), and **Week 6** (blue) post-extraction. Using this method, the dose–response curves for **Week 2** and **Week 4** exhibited substantial overlap, with H_50_ values of 47.3% (145.4 mOsm/kg) and 49.0% (150.6 mOsm/kg), respectively. No statistically significant difference was observed between these two time points (t = –2.06; *p* = 0.054). In contrast, significant differences in ED_50_ values were found between Week 2 and Week 6 (*p* < 0.0001), as well as between Week 4 and Week 6 (*p* = 0.0015).

**Figure 6 cells-14-01726-f006:**
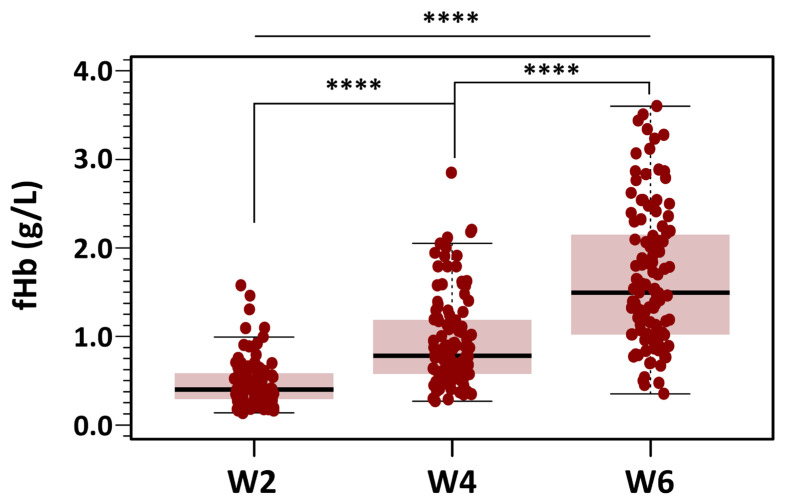
Changes in fHb concentration during RBC storage. Box plots display fHb values at different post-extraction days, grouped into three storage categories: days 3–14 (Week 2), days 18–28 (Week 4), and days 32–42 (Week 6). Mean fHb was 7.29 ± 4.34 µmol/L in the Week 2 group, 14.57 ± 7.91 µmol/L in the Week 4 group, and 25.73 ± 12.09 µmol/L in the Week 6 group. Results indicated statistically significant differences between the three groups with different storage time points (**** *p* < 0.0001 for all three groups).

**Figure 7 cells-14-01726-f007:**
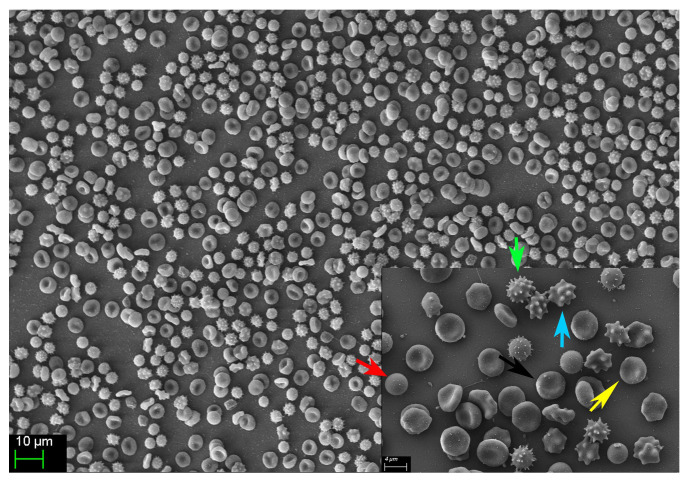
Representative image showing the diverse erythrocyte morphologies observed in one of the 20 analyzed pRBC units on day 42 post-collection. The normal, functional discocyte morphology (black arrow) is displayed alongside several abnormal, non-functional forms, including spherocytes (red arrow), crenocytes (green arrow), acanthocytes (blue arrow), and echinocytes (yellow arrow).

**Table 1 cells-14-01726-t001:** NaCl and PBS concentrations used to generate osmotic gradients for the assessment of RBC OF using Beutler’s spectrophotometric method, along with corresponding osmolality values (mOsm/Kg). This table presents the graded concentrations of NaCl (g/L) and equivalent percentages of PBS employed to create osmotic stress conditions for evaluating RBC membrane stability. These dilutions were applied according to Beutler’s spectrophotometric method. The corresponding osmolality values (mOsm/Kg) were calculated to quantify the osmotic load imposed at each step of the dilution series.

Osmolality (mOsm/Kg)	% PBS	Osmolality (mOsm/Kg)	[NaCl] (g/L)	Dilution
307.4	100	306.6	9.0	**1**
245.9	80	255.5	7.5	**2**
184.4	60	221.5	6.5	**3**
169.1	55	204.4	6.0	**4**
153.7	50	187.4	5.5	**5**
138.3	45	170.4	5.0	**6**
123.0	40	136.3	4.0	**7**
107.6	35	119.2	3.5	**8**
92.2	30	102.2	3.0	**9**
0.00	Distilled water	68.1	2.0	**10**
-	-	34.1	1.0	**11**
-	-	0.00	Distilled water	**12**

## Data Availability

This study was conducted according to the FAIR principles committed to making data and services Findable, Accessible, Interoperable, and Reusable (https://www.go-fair.org (accessed on 30 October 2025)). The entire anonymized dataset, metadata, and dictionary are available for download from https://zenodo.org (DOI:10.5281/zenodo.17086161).

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
