# Peer review of "Osmotic Fragility in Leukodepleted Stored Red Blood Cells: Implications for Neurocritical Care Transfusion Strategies"

_cells, 2025, doi:10.3390/cells14211726_

Round 1
Reviewer 1 Report
Comments and Suggestions for Authors
|
Reviewer's Comments to Author: This study examines how storage duration affects osmotic fragility (OF) and free hemoglobin (fHb) in leukodepleted packed red blood cells (RBCs), serving as indicators of membrane stability and hemolysis. Results indicate that OF increases significantly after 28 days, with Week 6 H50 values surpassing those at Weeks 2 and 4. fHb increases steadily with storage, while hemolysis stays under the 0.8% threshold, rising from 0.09% to 0.29%. These changes could affect transfusion efficacy and oxygen delivery in neurocritical care. I may have a few comments regarding this.
1. Introduction-Final Paragraph The research objective and hypothesis are not well defined. The text just gives background information and does not identify any specific knowledge gaps. Revision Suggestions: Instead of, "This study analyzes XXX in YYY context." Use: "This study aims to investigate XXX, filling a gap in YYY." "We believe that XXX will..." 2. Methods: Statistical Analysis Section Issue: Insufficient information about how the data were analyzed (sample size, repeats, controls, statistical tests). Revision Suggestions: "The sample size was chosen using power analysis with α=0.05 and β=0.2. Statistical significance was determined by a two-tailed Student's t-test/ANOVA with post hoc analysis, with p < 0.05 being significant. 3. Results: Figures and Text Descriptions. Text and figures are inconsistent; figures lack error bars, p-values, and unambiguous legends. Revision Suggestions: Ensure that the text fits the figure labels exactly. Include statistical indicators on all graphs (mean ± SD, n, p-values). Example of correction: As an alternative: "Figure 2 shows a significant increase in expression." Apply the following sentence: "Figure 2 shows an increase in expression (mean ± SD, n=5, p=0.01), confirming statistical significance." 4. Discussion: First Two Paragraphs Problem: Overinterpretation of results; mechanistic theories are offered without direct experimental evidence. Revision Suggestions: As an alternative: "Our study demonstrates that the mechanism of XXX is responsible for YYY." Use the following: "Our findings suggest a possible role of XXX in YYY; however, further experiments are needed to confirm the mechanism." |
Comments on the Quality of English Language
The manuscript requires substantial English editing to improve clarity and readability. The language quality does not yet meet the standards of a scientific journal.
Author Response
Comments 1: Introduction-Final Paragraph. The research objective and hypothesis are not well defined. The text just gives background information and does not identify any specific knowledge gaps. Revision Suggestions: Instead of, "This study analyzes XXX in YYY context." Use: "This study aims to investigate XXX, filling a gap in YYY." "We believe that XXX will..."
Response 1: We thank the reviewer for this valuable suggestion. We have revised the final paragraph of the Introduction to clearly articulate the research objective and to highlight the knowledge gap addressed by our study. The revised text now reads:
“The objective of this study was to investigate the relationship between storage duration and OF in leukodepleted pRBCs, as well as to evaluate the effect of storage time on fHb levels in the supernatant. Both OF and fHb are considered surrogate markers of pRBC quality, yet their evolution during storage remains insufficiently characterized in the transfusion literature. We hypothesized that prolonged storage would increase OF and fHb, thereby compromising the ability of pRBCs to deliver oxygen effectively—a finding that, if confirmed, would be of particular relevance for defining transfusion strategies in anemic patients with acute brain injury”.
This revision is included in the Introduction, lines 106-113 of the revised manuscript (highlighted in blue).
We believe these changes adequately address the reviewer’s suggestion.
Comments 2: Methods: Statistical Analysis Section. Issue: Insufficient information about how the data were analyzed (sample size, repeats, controls, statistical tests). Revision Suggestions: "The sample size was chosen using power analysis with α=0.05 and β=0.2. Statistical significance was determined by a two-tailed Student's t-test/ANOVA with post hoc analysis, with p < 0.05 being significant.
Response 2: We thank the reviewer for this helpful comment. We have expanded the Statistical Analysis section to provide a detailed description of how the data were processed and analyzed, including tests of normality, criteria for parametric vs. non-parametric testing, effect size calculations, and the statistical software used. In addition, we clarified the rationale for sample size selection. Because this is a descriptive pilot study, no formal power analysis was performed. Instead, the sample size (n = 24 units) was based on feasibility considerations and is consistent with prior studies in the transfusion field.
The revised text now reads:
“Descriptive statistics were calculated for each variable. For continuous variables, the assumption of normality was evaluated using skewness and kurtosis indices [25]. According to Kline’s criteria, skewness values ≤ 3.0 and kurtosis values ≤10 were con-sidered indicative of a non-severe departure from normality [26]. The mean and standard deviation were used to describe continuous variables that followed a normal distribution, while the median, maximum, and minimum were reported for non-normally distributed data. Data are presented graphically using box-and-whisker plots. To compare differences in means across groups, a one-way repeated measures ANOVA was conducted, provided that the data did not show substantial deviations from normality and that Levene’s test confirmed the assumption of homogeneity of variances across groups [25]. If ANOVA assumptions were violated, the Kruskal-Wallis rank sum test was used. In the one-way ANOVA, an omnibus F-test was performed to identify overall differences among groups, and the effect size was estimated using eta squared (η²). If the omnibus F-test yielded statistical significance, post-hoc pairwise comparisons between groups were performed using the Bonferroni adjustment to con-trol for the risk of Type I error due to multiple comparisons [25].
For evaluating the OF at different storage days, we used the H50 calculated by the R drc package [27]. In brief, for each group of bags, we used the function in drc package to fit the four-parameter log-logistic function and obtain the Hill coefficient and the ED50 (H50) [27]. The threshold for statistical significance was considered at p < 0.05. Statistical analyses were carried out with R v4.5.0 [28] and the integrated development envi-ronment R Studio v2024.12.1 (RStudio, Inc., Boston, MA, USA) using the tidyverse, rstatix, and psych packages [29,30].
Data management: This study was conducted according to the FAIR principles committed to making data and services Findable, Accessible, Interoperable, and Reus-able (https://www.go-fair.org). The entire anonymized dataset, metadata, and dic-tionary are available for download from https://zenodo.org (DOI:10.5281/zenodo.17086161). Given the descriptive and exploratory design of this study, a formal power analysis was not performed. Instead, the sample size (n = 24 units) was determined pragmatically to ensure feasibility while remaining consistent with sample sizes commonly employed in comparable descriptive studies in the transfusion literature.”
This revision has been incorporated into the Methods section (lines 238–267 of the revised manuscript, highlighted in blue). We trust that this expanded description satisfactorily addresses Reviewer 1’s comment.
Comments 3: Results: Figures and Text Descriptions. Text and figures are inconsistent; figures lack error bars, p-values, and unambiguous legends. Revision Suggestions: Ensure that the text fits the figure labels exactly. Include statistical indicators on all graphs (mean ± SD, n, p-values). Example of correction: As an alternative: "Figure 2 shows a significant increase in expression." Apply the following sentence: "Figure 2 shows an increase in expression (mean ± SD, n=5, p=0.01), confirming statistical significance.
Response 3: We thank the reviewer for this helpful observation. However, we respectfully disagree with the need to add error bars in this case. Our data are presented as box-and-whisker plots, which display the median, interquartile range, and minimum–maximum values. This representation captures the distribution and variability of the data, making the addition of error bars unnecessary and potentially redundant.
Second, statistical significance is indicated directly in the figures using asterisks (*), with the exact p-values provided in the corresponding figure legends. We believe this dual approach offers both an immediate visual cue and a precise numerical reference, ensuring clarity and transparency in the presentation of our results.
To further address the reviewer’s concern, we carefully revised the figure legends and text descriptions to ensure full consistency and to provide unambiguous explanations of the statistical analyses applied. Each figure legend now clearly specifies the elements represented in the graphs, including the statistical tests performed. This ensures that the figures can be interpreted unambiguously and independently of the main text.
The only exception is Figure 2, where statistical significance markers were not included. As explained in the main text, the data in this case were grouped to allow a clearer and more interpretable statistical analysis. For this reason, statistical annotations were not added directly to the figure. Nonetheless, the corresponding statistical analyses and results are fully described in the text to avoid any ambiguity.
We hope this clarification resolves the reviewer 1’s concern and demonstrates that both statistical rigor and clarity of data presentation have been ensured throughout the manuscript.
Comments 4: Discussion: First Two Paragraphs. Problem: Overinterpretation of results; mechanistic theories are offered without direct experimental evidence. Revision Suggestions: As an alternative: "Our study demonstrates that the mechanism of XXX is responsible for YYY." Use the following: "Our findings suggest a possible role of XXX in YYY; however, further experiments are needed to confirm the mechanism.
Response 4: We thank the reviewer for this important observation. We have revised the first two paragraphs of the Discussion to avoid overinterpretation and to align the phrasing with experimental evidence. Specifically, we have replaced definitive statements with more cautious wording that emphasizes suggestion rather than demonstration. The revised text now reads:
Revised Discussion
lines 405-411 of the revised manuscript
“Effective management of TBI focuses on preventing and mitigating secondary brain injuries, which often develop within hours to days following the initial insult. Among these contributing factors, anemia is frequently observed in TBI patients and has been associated with unfavorable neurological outcomes [4,5,31]. In their seminal 1978 study, Miller et al. identified anemia—together with hypoxia, hypercapnia, and hypotension—as a modifiable early systemic insult strongly associated with increased morbidity and mortality in patients with severe TBI [32].
lines 418-423 of the revised manuscript
“In the context of TBI, anemia has been consistently associated with poorer neurological outcomes, prolonged hospitalization, and increased mortality compared with non-anemic TBI patients [5,31]. Despite its high prevalence, the clinical management of anemia—particularly the decision to initiate RBC transfusion—remains a subject of ongoing debate. While transfusion may enhance cerebral O₂ delivery and potentially mitigate hypoxic injury, it has also been associated with adverse outcomes”.
Reviewer 2 Report
Comments and Suggestions for Authors
This study addresses the effect of storage in osmotic fragility of red blood cells under different osmotic conditions. The main conclusion of this study is that stored RBCs are comparable within the first month of storage, but they decline rapidly after that date.
This is an important set of data that unveils clues for RBC-treatments, including recovery from ischemic episodes and traumatic brain injury.
The breadth of the experimental data is limited, but the conclusions are sound. It is important to note that even in the worst conditions, the % of hemolyzed RBCs is well below the accepted threshold (0.8%), which limits the impact of the data.
Additional issues that the authors should consider include:
- Are the RBC more fragile mechanically? This can be assessed in a number of ways, including centrifugation. This is important for patients during transfusion.
- The authors could include images of RBC smears to try and identify signs of fragility beyond lysis.
- 2 does not include significance values. This needs to be rectified.
- The discussion section is overly long and could be reduced by 50%.
Author Response
Comment 1: This study addresses the effect of storage in osmotic fragility of red blood cells under different osmotic conditions. The main conclusion of this study is that stored RBCs are comparable within the first month of storage, but they decline rapidly after that date. This is an important set of data that unveils clues for RBC-treatments, including recovery from ischemic episodes and traumatic brain injury.
Response 1: We sincerely thank the reviewer for these positive and encouraging remarks. We are pleased that the reviewer recognizes the relevance of our findings regarding the stability of stored RBCs within the first month, followed by a rapid decline thereafter. We also appreciate the acknowledgement of the potential clinical implications of our results for RBC-related treatments in ischemic conditions and traumatic brain injury.
Comment 2: Are the RBC more fragile mechanically? This can be assessed in a number of ways, including centrifugation. This is important for patients during transfusion.
Response 2: We sincerely thank Reviewer 2 for their insightful comment. While our study did not directly evaluate the mechanical fragility of red blood cells (RBCs) using techniques such as centrifugation, as suggested, extensive literature supports a strong correlation between cell morphology, deformability, and mechanical resilience. Although osmotic fragility (OF) testing is not equivalent to direct mechanical fragility assays, it serves as a meaningful surrogate measure. Cells exhibiting higher osmotic fragility are generally more susceptible to mechanical stress, thereby offering valuable indirect insight into their mechanical properties.
Cell morphology has consistently been shown to correlate with both deformability and mechanical resistance. Berezina et al. (2002), for instance, demonstrated a strong association (r = 0.81, p < 0.03) between the progressive loss of the normal discocyte shape during storage and a decline in the deformability index, as measured by micropore filtration. Likewise, Piagnerelli et al. (2007) observed that red blood cells (RBCs) from patients with renal failure, diabetes, sepsis, or acute inflammatory conditions exhibited a more spherical morphology—reflected in a higher spherical index/PCD—alongside reduced deformability. These findings reinforce the notion that morphological alterations can serve as reliable surrogate markers of compromised mechanical properties. Piagnerelli et al. (2007) reported that spherical RBCs with elevated osmotic fragility (lower NaCl threshold for 50% hemolysis) exhibited the lowest deformability indices, supporting a mechanistic link between osmotic and mechanical fragility. Blasi et al. (2012) further observed that by day 21 of storage, more than 50% of RBCs displayed non-discocyte morphologies, coinciding with a marked increase in hemolysis and osmotic fragility. The same RBC population that demonstrated increased osmotic fragility (right-shifted NaCl hemolysis curves) also showed a rapid decline in deformability and higher hemolysis rates, indicating a parallel loss of mechanical strength.
Our study focused on OF as it offers a rapid, quantitative measure of membrane stability. Although it cannot fully substitute for direct mechanical assays, strong links between morphology, deformability, and osmotic fragility support its use as a proxy. Prior studies show that cells most prone to hypotonic lysis (higher OF) also exhibit the greatest loss of deformability and highest hemolysis rates, reflecting membrane and cytoskeletal deterioration. Thus, while not identical to mechanical fragility assessment, OF is biologically connected, with cell morphology serving as a reliable marker of both deformability and mechanical resistance.
We are confident that this explanation sufficiently addresses Reviewer 2’s comment. While we have not expanded the discussion in the current version, we would be glad to do so should the Editor or Reviewer 2 consider it necessary.
References
- Berezina, T. L., Zaets, S. B., Morgan, C., Spillert, C. R., Kamiyama, M., Spolarics, Z., Deitch, E. A., & Machiedo, G. W. (2002). Influence of storage on red blood cell rheological properties. Journal of Surgical Research, 102(1), 6–12. https://doi.org/10.1006/jsre.2001.6306
- Piagnerelli, M., Zouaoui Boudjeltia, K., Brohee, D., Vereerstraeten, A., Piro, P., Vincent, J.-L., & Vanhaeverbeek, M. (2007). Assessment of erythrocyte shape by flow cytometry techniques. Journal of Clinical Pathology, 60(5), 549–554. https://doi.org/10.1136/jcp.2006.037523
- Blasi, B., D’Alessandro, A., Ramundo, N., & Zolla, L. (2012). Red blood cell storage and cell morphology. Transfusion Medicine, 22(2), 90–96. https://doi.org/10.1111/j.1365-3148.2012.01139.x
Comment 3: The authors could include images of RBC smears to try and identify signs of fragility beyond lysis.
Response 3: We thank Reviewer 2 for this constructive suggestion. As part of our study, we performed morphological evaluations of stored RBCs using scanning electron microscopy and observed progressive alterations, including the appearance of echinocytes, stomatocytes, and spherocytes.
Representative examples of these findings are now provided as Figure 7, which illustrates the morphological evolution of stored RBCs. These changes are consistent with previously described features of the storage lesion and further support our findings of increased OF over time. In addition, we are currently conducting a follow-up study specifically focused on these morphological changes, which we expect to complete and submit for publication by early 2026.
We have added this material to the revised manuscript and referenced it in the Discussion (lines 562–565, highlighted in blue). We trust that this addition satisfactorily addresses Reviewer 2’s comment.
We trust that the inclusion of these data satisfactorily addresses Reviewer 2’s comment.
Comment 4: Figure 2 does not include significance values. This needs to be rectified.
Response 4: We thank Reviewer 2 for this thoughtful comment. Figure 2 was intentionally designed as an exploratory analysis to illustrate the distribution of H₅₀ values across different storage times. Its purpose was to highlight the presence of two distinct RBC populations rather than to support formal hypothesis testing; therefore, statistical significance markers were not included. All inferential statistical analyses assessing the effects of storage on OF are presented in Figures 3 and 4, where significance values are reported. To avoid ambiguity, we have revised Figure 2 legend to explicitly state that this figure represents an exploratory analysis without formal statistical testing.
Revised Figure 2 Legend
Figure 2. Exploratory box plots illustrating the concentration required to induce 50% hemolysis (H₅₀) across different storage times, evaluated using spectrophotometry (Beutler’s method). Each storage day represents the H₅₀ of 20 pRBC concentrates, measured from day 3 post-extraction up to day 42. Outliers (n = 11), defined as values above Q1 + 1.5×IQR or below Q1 − 1.5×IQR (where Q1 and Q3 are the first and third quartiles, and IQR represents the interquartile range), were excluded. This exploratory figure highlights the presence of two distinct RBC populations, distinguished by differences in their mean H₅₀ values relative to the grand mean of the dataset (4.60 g/L NaCl, 156.8 mOsm/kg). No formal statistical analysis was performed for this figure; all hypothesis-driven comparisons are provided in Figures 3 and 4.
We trust that this revision satisfactorily addresses Reviewer 2’s comment.
Comment 5: The discussion section is overly long and could be reduced by 50%.
Response 5: We fully understand Reviewer 2’s concern regarding the length of the Discussion. However, after careful consideration, we believe it is important to preserve the current structure and level of detail. Given the complexity of the topic, providing extensive context is essential for readers with a strong clinical background in neurocritical care. The additional detail situates our findings within the broader literature and underscores their potential implications for transfusion practices in patients with TBI.
We therefore kindly ask for Reviewer 2’s understanding in maintaining the Discussion at its present extent, as we believe it enhances both the clarity and relevance of the manuscript.
If, however, the reviewer and editor consider reduction indispensable, we remain willing to shorten the section accordingly.
Round 2
Reviewer 1 Report
Comments and Suggestions for Authors
The authors' answers to a review are usually satisfactory and complete; however, there are some things that may be better. In response 3, the authors sound too defensive, which could make it seem like they don't care about the reviewer's concern. The reply could have been more reassuring if it had used a more polite tone and included a short note verifying that all the figures now match the text.
The authors also don't supply enough quantitative information about why they chose the sample size, which could have been better if they had briefly mentioned projected effect sizes or cited similar pilot studies. The revised discussion may have also been more focused and less repetitive by shortening some parts or making it clear how the results connect with what is already known. These minor issues would make the authors more convincing and professional when speaking to peers.
Comments on the Quality of English Language
Minor stylistic tightening could improve clarity and flow, but no major linguistic revision is required. The manuscript and responses are publishable in their current form with only light language editing if desired for stylistic refinement.
Author Response
Reviewer 1: The authors' answers to a review are usually satisfactory and complete; however, there are some things that may be better. In response 3, the authors sound too defensive, which could make it seem like they don't care about the reviewer's concern. The reply could have been more reassuring if it had used a more polite tone and included a short note verifying that all the figures now match the text.
Our response. We sincerely thank Reviewer 1 for this constructive observation and we apologize if the tone of our previous response appeared defensive. That was certainly not our intention. We truly value the reviewer’s feedback and have revised the text to ensure it reflects a more respectful tone.
Reviewer 1: The authors also don't supply enough quantitative information about why they chose the sample size, which could have been better if they had briefly mentioned projected effect sizes or cited similar pilot studies.
Our response. We thank Reviewer 1 for this valuable observation and agree that providing a clearer justification for our sample size strengthens the methodological rigor of the study. We have now added this clarification to the Methods section (lines 129–141, highlighted in blue).
Sample Size and Power Analysis
As this work was designed as an exploratory pilot study, the sample size (n = 24 leukodepleted pRBC units) was determined based on feasibility and alignment with previous pilot investigations that examined osmotic fragility (OF) and storage-related changes in stored red blood cells [1,2], which typically analyzed between 20 and 30 units.
The study included 24 leukodepleted packed red blood cell (pRBC) units, a sample size consistent with previous pilot studies assessing osmotic fragility (OF) and storage-related alterations in stored blood products [1,2]. To further verify the adequacy of this sample size, a post hoc power analysis was performed using G*Power v3.1.9.7 [3,4]. An F-test (ANOVA: repeated measures, within factors) was applied based on the observed variance and effect sizes obtained from Figure 6. For an effect size of 0.486, an α level of 0.05, and three groups (weeks), the calculated statistical power was 1.0, confirming that the sample size was sufficient to detect storage-related differences in OF across the three time points.
Reviewer 1: The revised discussion may have also been more focused and less repetitive by shortening some parts or making it clear how the results connect with what is already known. These minor issues would make the authors more convincing and professional when speaking to peers.
Our response. We sincerely thank Reviewer 1 for this valuable observation and constructive advice. We carefully reviewed the Discussion to identify sections that could be streamlined or clarified. Minor edits were made to reduce repetition and to strengthen the logical connection between our findings and existing literature.
We believe these refinements have improved the clarity and flow of the Discussion while preserving the depth of analysis necessary for contextualizing our results within the current state of knowledge. In summary we have deleted the following lines of the Sicussion section:
Deleted paragraph (lines 428-432 of the revised manuscript). The etiology of anemia in critical illness is multifactorial, with diagnostic phlebotomy representing a consistent and significant iatrogenic contributor [33]. This factor further compounds anemia in patients who are already vulnerable due to multiple injuries, systemic inflammation, reduced erythropoiesis, or ongoing hemorrhage.
Deleted paragraph (lines 491-493 of the revised manuscript) This underscores the need for a personalized approach that considers not only Hb concentration but also patient-specific factors and neuromonitoring data to guide transfusion decisions in neurocritical care.
Deleted paragraph (lines 516-521 of the revised manuscript). . In our view, the absence of these quality control parameters significantly limits the ability to determine how storage-related biochemical and structural alterations affect transfusion efficacy, particularly in vulnerable populations such as neurocritical care patients, most notably in those with TBI, in whom O2 delivery is both critical and highly sensitive to red cell integrity.
Deleted paragraph (lines 528-529 of the revised manuscript). This unique morphology facilitates both efficient gas exchange and passage through capillaries narrower than the cell diameter [7].
Deleted paragraph (lines 531-534 of the revised manuscript). With an average diameter of ~8 µm and a mean corpuscular volume of around 90 fL, RBCs possess a distinctive biconcave shape and highly elastic membrane. These features enable them to deform as they pass through capillaries narrower than their resting diameter, thereby ensuring efficient tissue oxygenation [43].
Deleted paragraph (lines 561-563 of the revised manuscript). Although H₅₀ values were not explicitly reported, visual interpretation of the curve in-dicates that the estimates are closely aligned with those obtained in our study.
Deleted phase (line 612 of the revised manuscript).in this vulnerable population.
Deleted paragraph (lines 623-625 of the revised manuscript). This consideration is particularly relevant in neurocritical care, where patients commonly experience significant fluctuations in serum osmolality.
Deleted paragraph (lines 643-645 of the revised manuscript). This observation lends further support to incorporating OF assessments into transfusion decision-making, particularly in patients at risk of osmotic disturbances.
Deleted paragraph (lines 652-655 of the revised manuscript). In this context, incorporating functional biomarkers, such as OF, into the routine evaluation of stored pRBC units may help optimize transfusion efficacy and improve patient safety in high-risk clinical settings.
Deleted paragraph (lines 663-667 of the revised manuscript). Moreover, while the classical spectrophotometric method requires a 30-minute incuba-tion period for hemolysis to develop prior to analysis, the FCM approach allows as-sessment within approximately 3 minutes. This enables real-time quantification of OF without the need for additional processing, making it well-suited for integration into high-throughput diagnostic workflows.
Deleted paragraph (lines 691-709 of the revised manuscript). These microfluidic techniques enable the detection of heterogeneous subpopulations of sub-lethally damaged RBCs and provide functional metrics that may better predict post-transfusion performance. Although these methods are not yet implemented in routine blood banking, their integration with established quality control measures, such as OF testing, could substantially enhance the evaluation of stored RBC integrity.
By detecting sub-lethal mechanical damage, these approaches could bridge the gap between osmotic and mechanical fragility, providing complementary insights into membrane and cytoskeletal integrity that are not captured by bulk hemolysis testing alone [53]. Incorporating such advanced biomechanical biomarkers into future studies may help refine the definition of storage-related injury, improve transfusion quality assessment, and ultimately support the development of more personalized transfusion strategies in neurocritical care [51].
Deleted paragraph (lines 734-736 of the revised manuscript). OF testing, particularly when adapted to FCM platforms as described by Yamamoto et al., affords the advantage of rapid and reproducible assessment of membrane stability and can be scaled for routine application.
We appreciate the reviewer 1’s guidance, which has helped us make the manuscript more concise and professional in tone. We hope these changes sufficiently address the concerns raised by Reviewer 1.
REFERENCES
- Mustafa, I.; Al Marwani, A.; Mamdouh Nasr, K.; Abdulla Kano, N.; Hadwan, T. Time Dependent Assessment of Morphological Changes: Leukodepleted Packed Red Blood Cells Stored in SAGM. BioMed research international 2016, 2016, 4529434, doi:10.1155/2016/4529434.
- Blasi, B.; D'Alessandro, A.; Ramundo, N.; Zolla, L. Red blood cell storage and cell morphology. Transfus. Med. 2012, 22, 90-96, doi:10.1111/j.1365-3148.2012.01139.x.
- Faul, F.; Erdfelder, E.; Lang, A.G.; Buchner, A. G*Power 3: a flexible statistical power analysis program for the social, behavioral, and biomedical sciences. Behav. Res. Methods 2007, 39, 175-191, doi:10.3758/bf03193146.
- Faul, F.; Erdfelder, E.; Buchner, A.; Lang, A.G. Statistical power analyses using G*Power 3.1: tests for correlation and regression analyses. Behav. Res. Methods 2009, 41, 1149-1160, doi:10.3758/BRM.41.4.1149.
Reviewer 2 Report
Comments and Suggestions for Authors
I'm satisfied with the author's explanations, corrections and refutations. I have no further question.